# A Biosensor Platform for Metal Detection Based on Enhanced Green Fluorescent Protein

**DOI:** 10.3390/s19081846

**Published:** 2019-04-18

**Authors:** Woonwoo Lee, Hyojin Kim, Yerin Kang, Youngshim Lee, Youngdae Yoon

**Affiliations:** 1Department of Environmental Health Science, Konkuk University, 120 Neungdong-ro, Gwangjin-gu, Seoul 05029, Korea; lunia2005@hanmail.net (W.L.); gywls6772@naver.com (H.K.); yelin0514@naver.com (Y.K.); 2Division of Bioscience and Biotechnology, Bio/Molecular Informatics Center, Konkuk University, 120 Neungdong-ro, Gwangjin-gu, Seoul 05029, Korea; librashim@gmail.com

**Keywords:** biosensors, eGFP, heavy metal, split-protein systems

## Abstract

Microbial cell-based biosensors, which mostly rely on stress-responsive operons, have been widely developed to monitor environmental pollutants. Biosensors are usually more convenient and inexpensive than traditional instrumental analyses of environmental pollutants. However, the targets of biosensors are restricted by the limited number of genetic operon systems available. In this study, we demonstrated a novel strategy to overcome this limitation by engineering an enhanced green fluorescent protein (eGFP). It has been reported that combining two fragments of split-eGFP can form a native structure. Thus, we engineered new biosensors by inserting metal-binding loops (MBLs) between β-strands 9 and 10 of the eGFP, which then undergoes conformational changes upon interaction between the MBLs and targets, thereby emitting fluorescence. The two designed MLBs based on our previous study were employed as linkers between two fragments of eGFP. As a result, an *Escherichia coli* biosensor exhibited a fluorescent signal only when interacting with cadmium ions, revealing the prospect of a new biosensor for cadmium detection. Although this study is a starting stage for further developing biosensors, we believe that the proposed strategy can serve as basis to develop new biosensors to target various environmental pollutants.

## 1. Introduction

Environmental pollution has become a major problem in recent years and has been steadily increasing with rapid industrial development, adversely affecting, among other factors, human health. Several measures have been developed and applied to reduce environmental pollution. However, mitigating pollution remains a challenge because a massive amount of industrial materials are being released to the environment. To preserve environmental systems and relieve the adverse effects of pollutants, monitoring and detecting the amount of pollutants is an essential first step.

Environmental monitoring is typically focused on the quantification of target pollutants in the environment using analytical instruments [1,2]. Typical methods are time-consuming, expensive, and only provide the total amount of pollutants in a region. However, measuring the total amount of pollutants might lead to overestimation of their risk, as instrumental analysis usually extracts the general concentration of all pollutants from environmental samples but neglects pollutant–environment interactions [3,4]. Consequently, the amount of active pollutants affecting living organisms is usually below the total ratio of pollutants. In fact, the total amount and bioavailable portion of pollutants, especially in soils, have shown divergent results [5,6]. Therefore, bioavailability and portion of pollutants affecting living organisms should be considered to more accurately assess the risk of pollutants.

The bioavailability of pollutants can be determined considering living organisms such as fish, algae, plants, and animals as model systems [7,8,9,10,11]. In addition, the toxicity of pollutants can be determined by measuring diverse factors such as offspring, size and shape of organs, expression levels of markers (reporter genes), and growth rates. The variation of these factors can represent the toxicity of environmental pollutants. To overcome the difficulty to quantify the bioavailability of pollutants, microbial cell-based biosensors, also called whole-cell bioreporters (WCBs), have been recently developed [12,13,14]. WCBs are mainly based on stress-responsive operons in microbial cells. Exposing these cells to stress factors such as draughts, high temperatures, antibiotics, and metals makes the genes encoding the proteins to relieve stress by the interaction between the operon promoter and regulatory proteins [15,16]. Thus, WCBs can be generated by fusing the promoter region and reporter genes, such as fluorescent proteins or enzymes. In this case, the expression of reporter genes represents the exposure of bacterial cells to stress factors, and the expression levels are correlated to the amount of target material originating from these factors, thus enabling the quantification of target pollutants. Although WCBs have been successfully employed to quantify the bioavailable portion of pollutants, the number of stress-responsive operon systems for WCBs is limited with respect to the wide variety of environmental pollutants. 

Recently, various methods have been proposed to modulate target selectivity and specificity by biochemical and genetic engineering, and consequently overcome the limited number of available WCBs. Given that the expression of reporter genes in WCBs is determined by the interaction between target materials and regulatory proteins, the target specificity can be modulated by engineering regulatory proteins. In fact, it was shown that the selectivity and sensitivity of WCB based on zinc-responsive operon was modulated by replacing MBLs of ZntR, a regulatory protein in zinc-responsive operon, with different amino acid sequences in our previous reports [17,18,19]. Since amino acid sequences of MBLs of metalloregulators in metal-responsive operons including MerR, ZntR, GolS and ArsR known to interact with Hg, Cd, Au and As, respectively, were identified, it was possible to modulate the selectivity of MBLs by rational design [20,21,22]. Although it is possible to increase the number of WCBs to monitor a wider variety of pollutants, it would not be the most suitable solution. 

We propose a novel strategy to generate biosensors using fluorescent proteins as platform to target diverse environmental pollutants. Enhanced green fluorescent proteins (eGFPs) have been used in diverse biological studies given their high emission intensity [23,24]. For instance, eGFPs have been used as reporter genes to monitor the location of target proteins by N- or C-terminal tagging and as split-reporter proteins [25,26,27], in which two fragments of eGFP form an active fluorescent protein when associated. Based on this concept, we developed the biosensor platform using a split-eGFP system. To test our proposal, we chose two peptides sequenced as CNHEPGTVCPIC and CPGDDSADC modulated the selectivity of ZntR to cadmium and mercury, respectively [18]. Those two peptide sequences were inserted between β-strands 9 and 10 of eGFP as MBLs by replacing loop regions in the eGFP for it to be inactive. Then, the eGFP is activated by conformational changes induced by metal ion binding on the MBLs. Although many other factors such as binding constant and dissociation constant should be considered, the proposed strategy represents a promising and novel method to design biosensors. 

## 2. Materials and Methods

### 2.1. Materials and Instrumentation

*Escherichia coli* (*E. coli*) DH5α was used for gene cloning and *E. coli* BL21(DE3) for host strain of the WCBs and protein expression. The plasmids used in this study were pEGFP-N1 harboring eGFP gene and pCDFDuet (Merck KGaA, Darmstadt, Germany) for plasmid construction. CdCl_2_, K_2_Cr_2_O_7_, CuCl_2_·2H_2_O, HgCl_2_, NiCl_2_, AsCl_3_, PbCl_2_, AuCl_2_, SbCl_2_, and ZnCl_2_ were purchased from Sigma-Aldrich (St. Louis, MO, USA) and used to prepare 10 mM stock solutions. The fluorescence signals of eGFP were measured by FS-2 fluorescence spectrophotometer (Scinco, Seoul, Korea) equipped with a Xe lamp as a light source and bandwidth-adjustable filters for excitation and emission wavelengths.

### 2.2. Genetic Engineering of eGFP

The eGFP was engineered by inserting MBLs between the N-terminal region in 1–188 amino acids and the C-terminal region in 197–238 amino acids according to a previous report on a split-eGFP system divided between β-strands 9 and 10 [25]. The loop between β-strands 9 and 10 of the eGFP was replaced by MBLs known to interact with metal ions of cadmium, mercury, and zinc. Specifically, DNA sequences encoding the MBL inserted into eGFP by overlap extension polymerase chain reaction (PCR) replaced the loop between β-strands 9 and 10 [28,29]. First, the fragment encoding the N- and C-terminal regions were generated by PCR with the primers possessing the DNA sequences encoding the MLBs. Then, two fragments were used as template for a second PCR. As the end of both fragments contained the MBL region, they were overlapped and extended during PCR. The PCR products were inserted into pCDFDuet with *Bgl*II and *Not*I. The sequence of primers used in this study is listed in Table 1. The DNA sequences of the engineered eGFP with MBLs were confirmed by DNA sequencing. 

### 2.3. Characterization of Metal-Sensing Properties 

The WCBs were generated by introducing plasmid harboring engineered eGFP into *E. coli* BL21(DE3). To test the effect of metal ions, WCB cells were grown overnight at 37 °C and then inoculated into fresh media. Cell growth proceeded until the optical density at 600 nm (OD_600_) was around 0.4. I**s**opropyl β-D-thiogalactoside (IPTG) was added with 1 mM as final concentration to induce the expression of the engineered eGFP and incubated for 1 h. Then, the cells were exposed to 10 µM of diverse metal(loid) ions including arsenic, chrome, cadmium, nickel, mercury, lead, zinc, copper, gold and antimony. The cells were harvested every hour, and then their OD_600_ values and emission intensity of eGFP at 510 nm with excitation wavelength of 480 nm were measured. To measure the concentration-dependent responses toward the target metal ion, the WCBs were exposed to metal ion concentrations of 0–50 µM. The responses toward metal ions were first corrected by dividing fluorescence intensity with the OD_600_ values, because toxicity of the metal ions inhibited cell growth. Then, the responses were represented as induction coefficients defined as following equation: [response of WCBs with metal ion exposure]/[response of WCBs without metal ion exposure].

As the engineered eGFP in *E. coli* cells serves as metal sensor, the metal-sensing properties of the recombinant protein were also tested. To this end, WCB cells were grown overnight at 37 °C and then inoculated into fresh lysogeny broth media. Cell growth proceeded until OD_600_ reached 0.4. Then, protein expression was induced by adding 1 mM of i**s**opropyl β-D-thiogalactoside. The cells were grown overnight at 30 °C, harvested by centrifugation, and lysed by sonication with 50 mM Tris-HCl (pH 7.4) containing 160 mM NaCl. The recombinant proteins were purified by Ni-NTA resin (Qiagen, Hilden, Germany). To test their capability, the purified proteins were exposed to 10 µM of metal ions, and then the emission intensity of the engineered eGFP at 510 nm was measured every hour. Unlike the assay with whole cells, the intensity of eGFP was not corrected, but the responses toward metal ions were still represented as induction coefficients. 

### 2.4. Computational Evaluation 

Homology modeling of the engineered eGFP considered the X-ray crystallographic structure of eGFP (PDB ID: 4KA9) given its identical sequence and high-resolution determination at 1.58 Å. The engineered eGFP contained two different MBLs instead of four residues of eGFP, ^191^DGPV. Sequence alignment was carried out using the Clustal Omega software (http://www.ebi.ac.uk/Tools/msa/ clustalo/). Homology modeling was implemented on the Modeller 9v7 software developed by Andrej Sali (http://salibal.org/modeller/) based on the sequence alignment between a template eGFP and the MBL-inserted eGFP. The engineered eGFPs with inserted MBLs were further processed by energy minimization using the Sybyl 7.3 software (Tripos, St. Louis, MO, USA). All the generated eGFP structures were compared using the PyMol software developed by Warren Lyford DeLano (https://pymol.org/2/) and the UCSF Chimera software (https://www.cgl.ucsf.edu/chimera/).

## 3. Results

### 3.1. Genetic Engineering of eGFP for Biosensor

The MBLs were genetically introduced between the N- and C-terminal regions of eGFP. The MBLs reported in our previous study [18] replaced the four amino acids, DGPV, on the loop between N-terminal region in 1–190 amino acids and C-terminal region in 195–238 amino acids. The DNA sequences encoding MBLs were on the primers and introduced by two-step extension PCR (Table 1). 

The amino acid sequences for MBLs 1 and 2 were CNHEPGTVCPIC and CPGDDSADC, and known to prefer cadmium and mercury ions as ligands, respectively [18]. As the loop region was lengthened by inserting amino acids, the N- and C-terminal regions would not be associated to each other, thereby impeding fluorescence without metal ion treatment. However, when metal ions were associated to the MBLs, conformational changes were induced, and the two fragments were placed close. At a sufficiently short distance, the two fragments associate, and the engineered eGFP is fluorescent. This strategy was adopted in this study, and Figure 1 illustrates the mechanism of the engineered eGFP as metal biosensor.

### 3.2. Metal-Sensing Properties of Engineered eGFP in Whole E. Coli Cells

The WCBs were obtained by introducing plasmids carrying recombinant genes encoding the engineered eGFP into *E. coli* BL21. To determine the capability of engineered eGFP as biosensor for metal detection, the responses toward diverse metal(loid) ions were first tested. After expressing the engineered eGFP in *E. coli* cells by treating 1 mM of IPTG for 1 h, the cells were exposed to 10 µM of diverse metal(loid) ions, including arsenic, chrome, cadmium, nickel, mercury, lead, zinc, copper, gold, and antimony. The eGFP emission was measured at 510 nm with 480 nm for excitation over 1 and 2 h of exposure. The WCB responses were represented as induction coefficients. The engineered eGFP with loop 2 (eGFP-loop2) showed emission signals toward metal(loid) ions (Figure 2a), whereas the engineered eGFP with loop 1 (eGFP-loop1) showed no fluorescent signal (data not shown). Thus, we further investigated the eGFP-loop2.

Since the responses of WCBs were represented as induction coefficient values, it was not clear the strength of original signals. In order to clear this, we compared the fluorescence signals of WCBs harboring eGFP-loop 2, *E. coli* cells with and without reporter gene, *egfp*, in the presence of cadmium (Appendix A). The WCB without cadmium ion showed 180 arbitrary unit (AU) while with 5 µM of cadmium showed 460 AU. And *E. coli* cells with and without eGFP showed 1700 AU and 120 AU in the presence of 5 µM of cadmium, respectively. This suggests that the insertion of MBLs in this study effectively inactivates the eGFP. On the other hand, the emission signals are the highest with metal ions in the following order: cadmium, mercury, and nickel. The metal specificity of the proposed biosensor is discussed below. The emission signals also suggest that the metal ion–MBL interaction induces conformational changes in the engineered eGFP to be active. 

Metal ion selectivity was further investigated with the WCB harboring eGFP-loop2 given its sensitivity to cadmium, mercury, and nickel. Specifically, the WCB was exposed to different concentrations of those three metal(loid)s. Figure 2b shows that only the Cd emission increases according to concentration. In fact, the responses of eGFP-loop2 to Ni and Hg did not correspond to eGFP signals but are more likely associated with the representation of emission signals as induction coefficients. As the toxic effects of metal ions were corrected based on the OD_600_ values, the high toxicity of Ni and Hg appeared to retrieve considerable responding signals. However, the concentration tests revealed that the generated WCB based on genetic engineering of eGFP only has cadmium selectivity.

### 3.3. Specificity of WCBs to Cadmium Ions

We investigated the specificity of WCBs to metal(loid) ions according to concentration. Concentrations from 0 to 5 µM of metal(loid) ions were used to verify specificity and reduce the influence of metal(loid) toxicity. We followed the experimental procedure described in previous sections while varying the concentration of metal(loid) ions. Concentration-dependent responses were not observed using Hg and Ni, and hence, these two metal(loid) ions were excluded from further analysis (data not shown). In contrast, the emission signals of WCB to Cd increase with concentration, as shown in Figure 3a. The emission intensity of eGFP induced by Cd ions, represented as induction coefficients, reached 2.5 at 5 µM. This increase in the emission signal, however, is not substantial, especially compared to other WCBs based on stress-responsive operons. Still, it can be used for quantitative analyses of cadmium, because the emission–concentration relation has a coefficient of determination *R*^2^ = 0.987 obtained from linear regression (Figure 3b). This result suggests that the WCB harboring eGFP-loop2 can be a suitable prospect of cadmium biosensor. 

### 3.4. Metal-Sensing Properties of Recombinant Engineered eGFP 

The eGFP-loop2 acting as cadmium biosensor in *E. coli* cells led us to speculate that purified proteins might work as biosensors. To test this idea, recombinant proteins were purified from *E. coli* cells using Ni-NTA resin. Then, 10 µM of metal(loid) ions were treated to diverse concentrations of protein solutions, and the emission of eGFP at 510 nm was measured at 480 nm of excitation wavelength. The emission was measured every hour during 6 h after addition of metal ions, but no emission signal was detected (data not shown). If the engineered eGFP were stable after purification, the same result obtained from the whole-cell assay would be expected. However, no difference occurred in the emission intensity with and without metal(loid) ion treatment. These results suggest that the engineered eGFP with insertion of additional amino acid sequences was not stable enough to constitute a biosensor. In fact, the split-eGFP has been reported as not being very stable, and stabilizing the protein by introducing point mutations has been attempted [26]. Although the engineered eGFP developed in this study did not behave as a biosensor, further investigations should be conducted.

### 3.5. Computational Analysis of Engineered eGFP

To understand the mechanism of the biosensors for metal detection developed in this study, we analyzed their three-dimensional structures. As the structure of the engineered eGFP was not available, we implemented one using Modeller 9v7 based on the X-ray crystallographic structure of the eGFP deposited in the Protein Data Bank (https://www.rcsb.org/). The structures were obtained from Modeller using amino acid sequence alignments and then subject to energy minimization using Sybyl 9.3. Three residues, namely, 65T, 66Y, and 67G, have been biosynthesized into chromophore in the X-ray crystallographic structure. Hence, we generated a wild type model structure using Modeller for comparison with the modified eGFPs. The wild type model structure was almost identical to the X-ray crystallographic structure (4ka9.pdb), except for the chromophore region and loops. The root-mean-square error between the template and model structure was 0.8. The structures of wild type eGFP and the two engineered eGFPs are shown in Figure 4. The amino acid sequence of loops 1 and 2, CNHEPGTVCPIC and CPGDDSADC, respectively, were inserted instead of DGPV located at the loop between β-strands 9 and 10. As a result, the structures were similar, but the chromophore motif of eGFP consisting of Thr65, Tyr66, and Gly67 changed in the engineered eGFP. By comparing the structures with loops 1 and 2, the direction of helical turn changes (Figure 4d), and the turn of chromophore in eGFP-loop1 is different from that in the other eGFPs. To understand how the three residues of eGFP-loop1 do not form the chromophore, the Phi and Psi angles of this region were examined. The Ramachandran plots of the structures are shown in Figure 5. In Figure 5a, the residues in the chromophore show positive Phi angles, which are unusual for the normal α-helix structure. Such unusual angles may be explained by the cyclization of 65T and 67G occurring during chromophore biosynthesis. However, the negative Phi angles of 66Y and 67G were observed in the modified eGFP-loop1 (Figure 5b) as values of −48.2 and −83, respectively. The modified eGFP-loop2 (Figure 5c) showed a negative Phi angle of 67G, and two positive angles of 65T and 66Y. 

From the results, it is hard to explain why the engineered eGFP is fluorescent with metal ions. However, we speculate that the metal ion binding on the MBLs induced conformational changes, and the two parts of the eGFP approached, thereby activating the protein. 

## 4. Discussion

Microbial cell-based biosensors, also called WCBs, have been actively developed and investigated during the past few decades given their advantages over traditional instrumental analysis. However, WCBs have not been frequently applied to environmental monitoring by the limited number of possible targets and broad selectivity. This limitation arises from most WCBs being based on stress-responsive genetic systems. Moreover, the WCBs should be modified to improve specificity to target pollutants, as living organisms have different signaling pathways. Therefore, it would be difficult to target various environmental pollutants and perform quantitative measurements with conventional WCBs. 

To compensate the shortcomings of WCBs, we tried to modify the regulatory proteins involved in operons. In a previous study, we reported a cadmium-sensing WCB based on the znt operon, zinc inducible operon, that responds to cadmium ions by their interaction with ZntR, which is a regulatory protein of this operon [30]. Given that the cadmium response was induced by the ZntR–cadmium ion interaction, it would be possible to modify the metal ion specificity of WCBs by changing the MLBs of ZntR. As reported previously, the specificity of WCBs toward metal ions can be modified by genetic and biochemical engineering [18,19]. However, this approach does not notably increase the variety of target pollutants, because the metal-binding regions of the regulatory proteins involved in other genetic operons, such as ArsR, CueR, NikR, and TetR in arsenic-, copper-, nickel-, and tetracycline-inducible operons, respectively, consist of several amino acids that are not close at the level of primary structure. Therefore, we developed a novel strategy to generate biosensors based on the concept of split-eGFP. 

Split-protein systems are frequently used to measure either protein–protein or ligand–protein interactions. Hence, they may be applied to detect environmental pollutants [27]. The biosensors demonstrated in this study were based on the eGFP and structurally modified by genetic engineering. It has been reported that the split-eGFP divided into N- and C-terminal regions can form mature proteins when two fragments are associated [26]. Therefore, we used the split-eGFP system to generate new biosensors for metal ion detection. Figure 1 shows the two speculations included in our experimental design. First, the separated N- and C-terminal regions of the engineered eGFP with MBLs is not fluorescent. Second, binding metal ions on the MBLs induces conformational changes on the engineered eGFP to associate the two fragments, thus emitting fluorescent signals. 

To evaluate these aspects, two peptides sequenced as CNHEPGTVCPIC and CPGDDSADC were selected as MBLs and inserted into the eGFP. We selected these two peptides because they showed specificity to cadmium and mercury when replacing the MBL in ZntR from our previous study [18]. When the engineered eGFPs were overexpressed in *E. coli* without metal ion treatment, there was no green fluorescent signal. This result verified our first speculation, because the wild type eGFP is fluorescent (Figure 2). The second speculation was also verified because the engineered eGFP-loop2 showed fluorescent signals in response to cadmium ions, whereas the eGFP-loop1 showed no fluorescence (Figure 2). Hence, the MBL with amino acid sequence CPGDDSADC may interact with cadmium ions and induce the conformational changes to associate the two fragments of eGFP. Nonetheless, it was noticed that the induction coefficient values from eGFP-loop2 was relatively weak compared to WCB employing *znt*-operon regulated by ZntR in our previous reports [18]. In fact, it was natural because these two systems as metal-sensing biosensors were different. One employed native eGFP as a reporter and other employed inactivated eGFP as a sensor molecule that was known as instable as mentioned in the Results section [26]. Thus, it would be possible to enhance the sensitivity of our new sensors by improving the stability of engineered eGFP. 

Additionally, we should consider the unresponsiveness of eGFP-loop1 to metal ions. When the MBL in ZntR was replaced by loop 1, it interacted with cadmium ions to induce the expression of the reporter gene. To verify this fact, the amino acid sequences of both peptides were analyzed, and it was clear the length was different. As the length of the loop region is related to protein stability, a longer loop may have hampered the association of the two fragments of eGFP. To elucidate this assumption, the three-dimensional structures of the engineered eGFPs were built using Modeller based on the structure of eGFP (Figure 4). The residues comprising chromophore were twisted in distinct ways in the model structure of eGFP-loop1, possibly disabling the chromophore. The modified loop was placed beside the fourth helix connected to the chromophore, but the longer loop may compromise interaction with this helix, and the chromophore may be unstable. Of course, these results cannot fully explain why only eGFP-loop2 worked as biosensor but provide insights to improve our strategy for developing biosensors. 

The detection of environmental harmful materials can be a first step to improve life sustainability. The monitoring of well-known pollutants such as chemicals and heavy metals are relatively well-established. However, simpler and faster methods are required to quantify diverse environmental pollutants and overcome the shortcomings of traditional instrumental analysis. Thus, we believe that the development of novel biosensors will be increasingly important and propose a strategy to develop biosensors for metal detection based on split-eGFP, conforming the first attempt to develop such biosensors using split-protein systems. Although the proposed strategy requires further improvements and considerations to enable the proper design of biosensors, it can be very valuable to fields related to environmental monitoring for increasing the diversity and capabilities of biosensors.

## Figures and Tables

**Figure 1 sensors-19-01846-f001:**
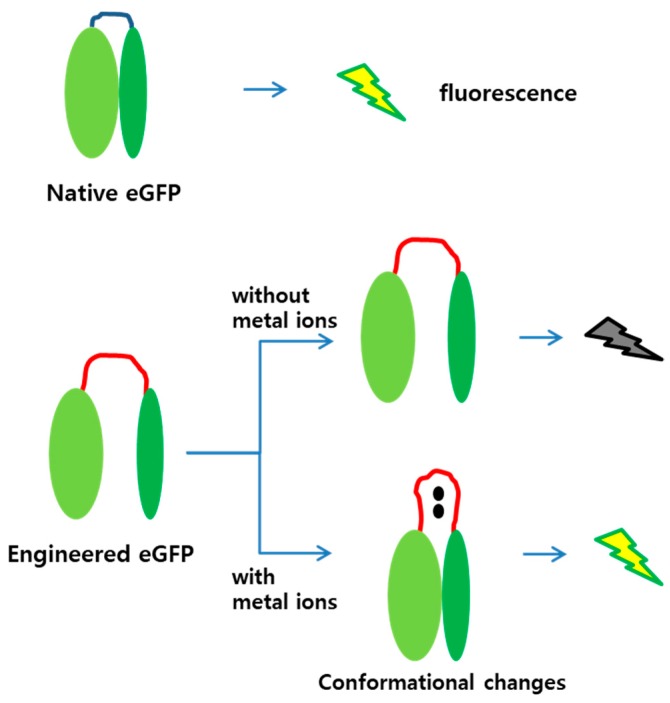
Diagram of mechanism for engineered eGFP as sensing molecule. The eGFP is active in its native form, whereas the engineered eGFP is inactive due to loop insertion. When a metal ion binds to the MBL, the two parts of the engineered eGFP approach each other and associate to become active. (Red lines, MBL inserted into the eGFP; black dots, metal ions).

**Figure 2 sensors-19-01846-f002:**
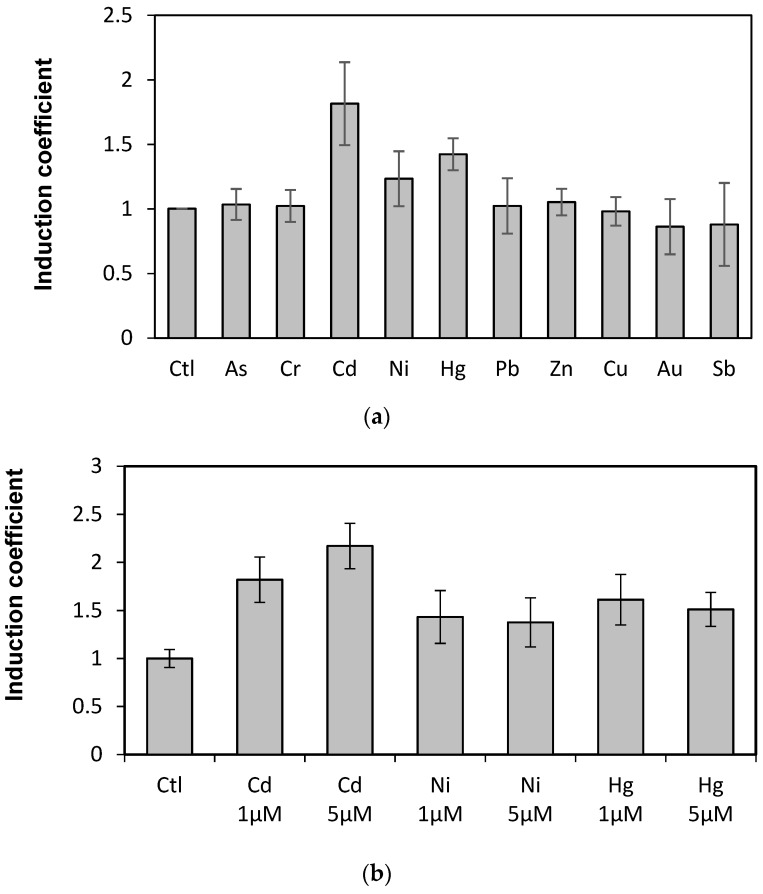
Induction coefficients from *E. coli* cells harboring eGFP-loop2 exposed to metal ions. (**a**) Responses of engineered eGFP-loop2 to various metal(loid) ions. (**b**) Induction coefficients of eGFP-loop2 to different concentrations of Cd, Ni, and Hg. (Ctl, control without metal ions).

**Figure 3 sensors-19-01846-f003:**
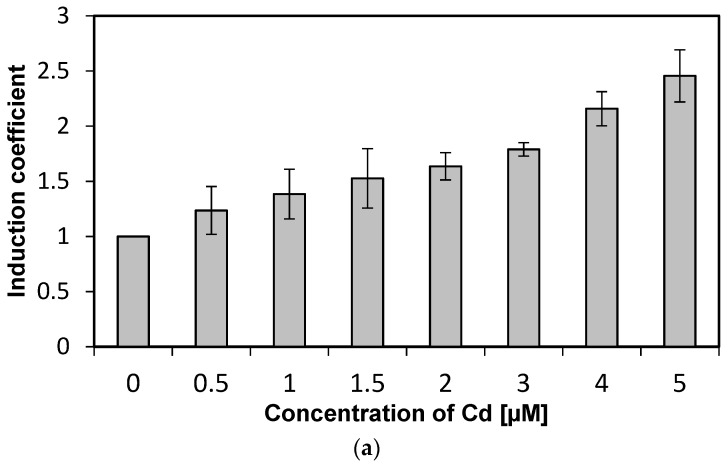
Concentration-dependent responses of *E. coli* cells with eGFP-loop 2 to Cd ions. (**a**) Induction coefficients for Cd concentrations from 0 to 5 µM. (**b**) Linear regression of Cd induction coefficient according to concentration. (*y*, induction coefficient; *x*, concentration of Cd; *R*^2^, coefficient of determination from linear regression).

**Figure 4 sensors-19-01846-f004:**
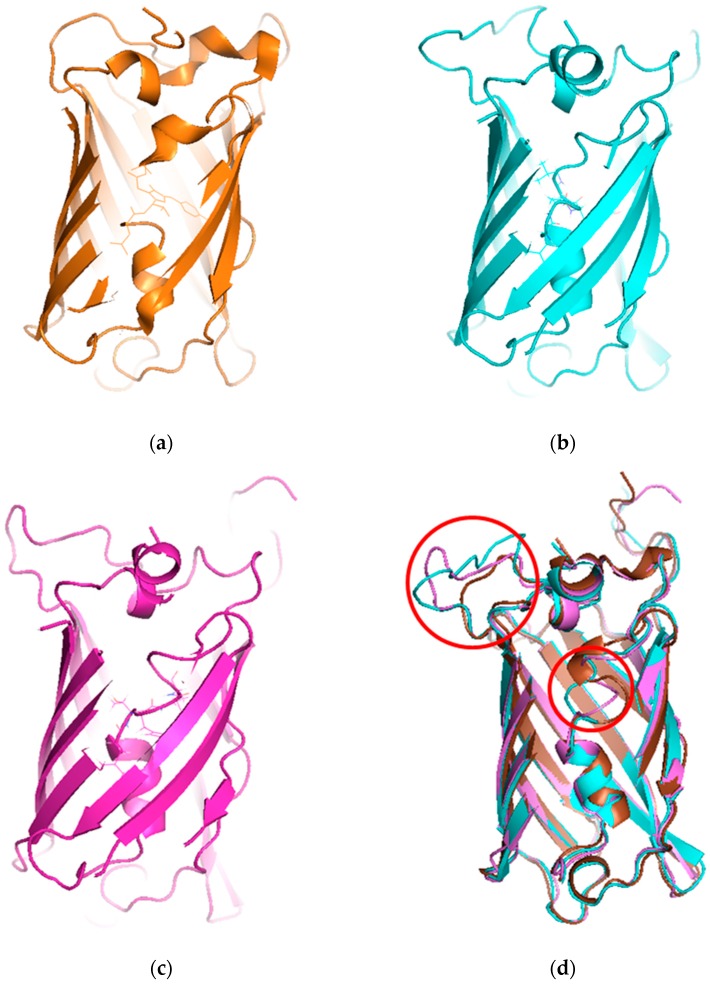
Structures of X-ray crystallographic eGFP and model structures with modifications. (**a**) Wild type structure of eGFP (4ka9.pdb). (**b**) Model structure of eGFP-loop1 (CNHEPGTVCPIC). (**c**) Model structure of eGFP-loop2 (CPGDDSADC). (**d**) Superimposed model of the three structures (modified loops and chromophore are enclosed in red circles).

**Figure 5 sensors-19-01846-f005:**
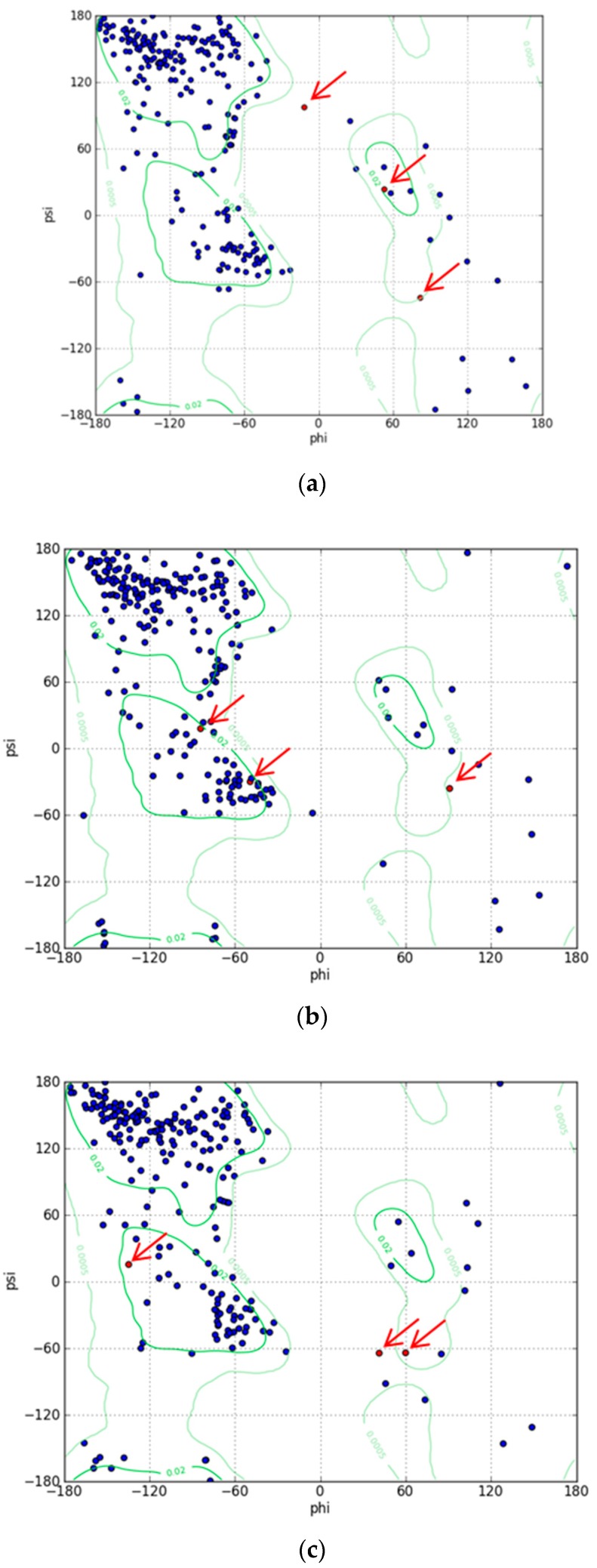
Ramachandran plots for model structures. (**a**) Wild type structure of eGFP. (**b**) Structure of eGFP-loop1. (**c**) Structure of eGFP-loop2. (The three residues, 65T, 66Y, and 67G, in the chromophore are indicated with red arrows).

**Table 1 sensors-19-01846-t001:** Primers and amino acid sequences for MBLs used in this study.

Primer	Sequence (5’ to 3’)	Restriction Enzyme Site	Amino Acid Sequence
1	GT***AGATCT*****C**ATGGTGAGCAAGGGCGAG	*Bgl*II	
2	AT***GCGGCCGC***CTTGTACAGCTCGTCCATGC	*Not*I	
3	*TGCAACCATGAACCGGGCACCGTGTGCCCGATTTGC*CTGCTGCCCGACAACCACTAC	–	CNHEPGTVCPIC
4	*GCAAATCGGGCACACGGTGCCCGGTTCATGGTTGCA*GCCGATGGGGGTGTTCTGCTG	–	
5	*TGCCCTGGCGATGACAGCGCCGACTGC*CTGCTGCCCGACAACCACTAC	–	CPGDDSADC
6	*GCAGTCGGCGCTGTCATCGCCAGGGCA*GCCGATGGGGGTGTTCTGCTG	–	

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
