# Peer review of "A Biosensor Platform for Metal Detection Based on Enhanced Green Fluorescent Protein"

_sensors, 2019, doi:10.3390/s19081846_

Reviewer 1 Report

The paper submitted by Y. Yoon presents a proof of concept based on enhanced green fluorescent proteins sensor for heavy metal detection.  The proposed strategy represents an innovative and novel strategy for biosensors design.

The manuscript is well organized and the conclusions are sustained by the experiments. The paper should be accepted after answering the questions:

1. How the authors explain the selectivity for cadmium?

2. In the experimental part please add information about the metal ions solution used in this study.

3. Why the authors chosen 10 μM concentration for metal ions?

Author Response

Reviewer 1

Comments and Suggestions for Authors

The paper submitted by Y. Yoon presents a proof of concept based on enhanced green fluorescent proteins sensor for heavy metal detection.  The proposed strategy represents an innovative and novel strategy for biosensors design.

We appreciate your thoughtful comments. We addressed the point-by-point explanations for comments raised by reviewer below and made appropriate changes in the revised MS. 

The manuscript is well organized and the conclusions are sustained by the experiments. The paper should be accepted after answering the questions:

1. How the authors explain the selectivity for cadmium?

Thanks for your comment on this issue. As described in the MS, the WCB with loop 2 showed best response toward Cd and weak positive signals with Hg and Ni from selectivity test (Fig. 2a). To clear this issue, the concentration dependent test was followed with 3 metal(loid) ions (Fig. 2b). As a result, the new sensor did not showed concentration dependent responses toward any metal(loid) ions except Cd. Thus, we concluded the new sensor has Cd selectivity.

At this point, we understood the reviewer’s concern about this conclusion because the sensor still showed positive signals toward Ni and Hg as shown in Fig. 2b. However, we thought it would be caused by the toxic effects of metal ions and the way we calculated the induction coefficient values as we described in the MS. In fact, we tested with higher concentration ranges of Ni and Hg, but the WCB did not show significant fluorescent signals.

2. In the experimental part please add information about the metal ions solution used in this study.

It was our mistake. The information about metal ions was now included in Materials and Methods section in the revised MS.

3. Why the authors chosen 10 μM concentration for metal ions?

             We tested diverse concentration of metal ions to screen the selectivity. The concentration less than 10 μM showed relatively weak signal and higher showed strong toxicity on E. coli cells. In these reasons, we decided to 10 μM for selectivity tests because the WCBs showed clear differences and moderate toxicity at 10 μM of metal ions.

Reviewer 2 Report

Overall an interesting study even with some negative data including non-functional loop 1 and non-functional purified protein, but the detailed and logical discussion complemented the results really well.

Introduction:

·         Need a bit more background about the two peptides and why there were chosen for this study in the Introduction section. In its current form, this information was only revealed in the second paragraph of the Discussion section, leaving readers confused reading through the methods and results.

·         Since the foundation of this manuscript is metal-binding, the introduction should include relevant literatures about how metals interaction with the regulatory protein of the metal-resistance operon.

·         Target inducible promoter-reporter gene fusion can be talked about less in the introduction since that’s not how your biosensor platform is built.

Methods:

·         Information on the specific chemicals used in the study is missing. For example, was cadmium chloride used in this study? What form of arsenic was used? etc.

·         What instrument was used to measure fluorescence?

·         Please include equations (or something like that) to clearly indicate how fluorescence was corrected by OD values and how induction coefficient was calculated.

Results:

·         There was no direct data proving that the engineered eGFP-loop2 was not fluorescent without metal treatment. A side-by-side comparison of wild type E. coli (negative control), E. coli expressing the engineered eGFP, and E. coli expressing the native eGFP (positive control) will be more appropriate to demonstrate this point.

Discussion:

·         The overall induction seemed to be weak with the highest induction coefficient being 2.5. How did this compare to your previous study (ref 19)? A discussion regarding this and maybe suggestions on how to improve induction would be helpful.

Author Response

Reviewer 2

Overall an interesting study even with some negative data including non-functional loop 1 and non-functional purified protein, but the detailed and logical discussion complemented the results really well.

We appreciate your constructive criticism. The issues pointed out were addressed below and the appropriate changes were made in the revised MS.

Introduction:

1. Need a bit more background about the two peptides and why there were chosen for this study in the Introduction section. In its current form, this information was only revealed in the second paragraph of the Discussion section, leaving readers confused reading through the methods and results.

             Thanks for your critical comment on this issue. In this study, we focused on whether our proposal was going to work or not. Thus, we needed a peptide possessing a potential to bind metal ions. In our previous study, we tried to generate metal sensing WCBs by changing MBL in ZntR, and those two selected peptides modulated the original metal selectivity of ZntR. It was the reason we chose this, and the information was included in the revised MS.

2. Since the foundation of this manuscript is metal-binding, the introduction should include relevant literatures about how metals interaction with the regulatory protein of the metal-resistance operon. 

           The cited references about WCBs did not show the structural analysis between MBLs and metal ions. As following reviewer’s suggestion, we added other references describing the interaction between MBLs and metal ions in the Introduction.

3. Target inducible promoter-reporter gene fusion can be talked about less in the introduction since that’s not how your biosensor platform is built.

             Thanks for your suggestion. However, we thought it was acceptable because current stage of biosensors was on the fusion of promoter and reporter gene. As you pointed out, our approach was not related to this much, but it would be necessary to explain the background of biosensors.

Methods:

1. Information on the specific chemicals used in the study is missing. For example, was cadmium chloride used in this study? What form of arsenic was used? etc.

 It was a mistake. The information was now included in the revised MS.

2. What instrument was used to measure fluorescence?

We used FS-2 fluorescence spectrophotometer (Scinco, Seoul, Korea) equipped with a Xe lamp as a light source and bandwidth-adjustable filters for excitation and emission wavelengths. This information was included in the revised MS.

3. Please include equations (or something like that) to clearly indicate how fluorescence was corrected by OD values and how induction coefficient was calculated.

In this study, we measured the fluorescence from 1 mL of E. coli cells. The cells were exposed to metal ions to induce the signals, the toxic effects slowed down the growth of cells. In this reason, it would not be acceptable to measure only the fluorescence signals. Thus, we corrected the fluorescence signals by dividing with OD600 values. Of course, it could not rule out the toxicity issues, but we believed it was reasonable. After correct the signals, the induction coefficient values were calculated by following equation defined as [fluorescence intensity of heavy metal–exposed biosensor]/[fluorescence intensity of non-exposed biosensor].

             It was added in Materials and Methods section in the revised MS.

Results:

1. There was no direct data proving that the engineered eGFP-loop2 was not fluorescent without metal treatment. A side-by-side comparison of wild type E. coli (negative control), E. coli expressing the engineered eGFP, and E. coli expressing the native eGFP (positive control) will be more appropriate to demonstrate this point.

             We appreciate your constructive comment. In fact, it had been tested before we performed this study, but it was not included in the MS. In case of negative and positive controls, the fluorescence signals at 1 hr after IPTG treatment were 120 AU (arbitrary unit) and 1760 AU, respectively. The signal from E. coli w/o eGFP was background signal even if it was varied upon cell density. In case of eGFP-loop2, the signals were 180 AU and 460 AU for without and with metal ion, respectively. Thus, the induction coefficient values were about 2.5 for metal treatment. It would be clearly shown in figures below.

             We thought it was not necessary to be included in the MS, but we could add to revised MS if it needed.

Discussion:

1. The overall induction seemed to be weak with the highest induction coefficient being 2.5. How did this compare to your previous study (ref 19)? A discussion regarding this and maybe suggestions on how to improve induction would be helpful.

             Thanks for your thoughtful comment on this. As you pointed out, the induction coefficient values from eGFP-loop2 was much less than that of WCB employing znt promoter and egfp as reporter gene. It was natural because one used native eGFP as reporter and the other used inactivated eGFP-loop2 as sensor molecule. In case of native eGFP, all expressed proteins were fluorescent, while only a portion of eGFP-loop2 was fluorescent. In addition, the stability of split-eGFP was less than the native eGFP.

             To enhance the sensitivity of our sensor, it was necessary to improve the stability by introducing point mutations based on structural analysis. It would be future goal of our study and should be answered to use our system as platform for biosensors.

             The additional discussion was included in the revised MS.

Round  2

Reviewer 2 Report

Thank you for your thorough response. They adequately addressed my concerns. I would still recommend the figure showing the comparison of eGFP-loop2 and native eGFP (the one you included in the response) to be included in the manuscript, as it is still good to know the overall fluroescence intensity of the engineered protein. 

Author Response

Thanks for your comment.

We agreed to your suggestion and the figure showing the comparison of orignal fluorescence signal from eGFP-loop2, E. coli with and without  native GFP in the presence of cadmium is now included in the revised MS as supplementary figure. The explanation about this was addresed in the Results section.